# Drought Forecasting: A Review and Assessment of the Hybrid Techniques and Data Pre-Processing



**Mustafa A. Alawsi** [1,2], **Salah L. Zubaidi** [2], **Nabeel Saleem Saad Al-Bdairi** [2], **Nadhir Al-Ansari** [3,*] and **Khalid Hashim** [4,5]

1   Department of Building and Construction Techniques-Kut Technical Institute, Middle Technical University, Wasit 52001, Iraq; mustafa.abdulgani@mtu.edu.iq
2   Department of Civil Engineering, Wasit University, Wasit 52001, Iraq; salahlafta@uowasit.edu.iq (S.L.Z.); nsaleem@uowasit.edu.iq (N.S.S.A.-B.)
3   Department of Civil Environmental and Natural Resources Engineering, Lulea University of Technology, 971 87 Lulea, Sweden
4   Built Environment and Sustainable Technologies (BEST) Research Institute, Liverpool John Moores University, Liverpool L3 3AF, UK; k.s.hashim@ljmu.ac.uk
5   Department of Environment Engineering, Babylon University, Babylon 51001, Iraq
*   Correspondence: nadhir.alansari@ltu.se

**Abstract:** Drought is a prolonged period of low precipitation that negatively impacts agriculture, animals, and people. Over the last decades, gradual changes in drought indices have been observed. Therefore, understanding and forecasting drought is essential to avoid its economic impacts and appropriate water resource planning and management. This paper presents a recent literature review, including a brief description of data pre-processing, data-driven modelling strategies (i.e., univariate or multivariate), machine learning algorithms (i.e., advantages and disadvantages), hybrid models, and performance metrics. Combining various prediction methods to create efficient hybrid models has become the most popular use in recent years. Accordingly, hybrid models have been increasingly used for predicting drought. As such, these models will be extensively reviewed, including preprocessing-based hybrid models, parameter optimisation-based hybrid models, and hybridisation of components combination-based with preprocessing-based hybrid models. In addition, using statistical criteria, such as RMSE, MAE, NSE, MPE, SI, BIC, AIC, and AAD, is essential to evaluate the performance of the models.

**Keywords:** data pre-processing; drought; hybrid models; machine learning; performance metrics

## 1. Introduction

Drought has become more common all over the world in the last few decades because of climate change and global warming [1]. It is one of the most expensive and least understood natural disasters. It may have devastating consequences on agriculture, water supply, ecosystems, public health, watershed health, and the economy. Each year, the United States suffers losses of six to eight billion dollars as a result of droughts [2]. Yan et al. [3] mentioned that drought is also a major natural disaster in China, with its frequency reaching 70% in certain places during the summer. Moreover, it causes enormous socioeconomic losses, especially in agriculture. According to the United Nations (2011), drought conditions in the Horn of Africa in 2011 resulted in significant starvation in Kenya, Ethiopia, Djibouti, and Somalia [4]. Therefore, it is defined as a complex and poorly understood phenomenon because of the complexity of its contributing factors and its underlying driven factors [5]. It is usually caused by a reduction in rainfall over a long time [6]. Furthermore, in some cases, it occurs due to abnormalities in evapotranspiration and temperature [7]. As a result, it has a long-lasting effect over broad regions, lasting months or even years. It affects food production and has detrimental impacts on the economic performance of large areas or whole nations [8,9].

Climate change has a considerable challenge for human beings in the future since it has the potential to cause more radical global occurrences with enormous socio-economic consequences [10,11]. As a result of growing international attention from governments and scholars, many resources have been devoted to developing measures to mitigate the effects of climate change and reverse its consequences. It is projected that the impact of climate change will likely increase in the future due to the global warming effect, which will increase the evaporation of water. This issue will most likely result in more acute drought occurrences [12,13]. Additionally, climate change affects the ecology in several ways, including but not limited to, changing precipitation patterns that may cause drought and desertification. Moreover, climate change threatens human lives and livelihoods with several consequences (i.e., depletion of freshwater supplies, population growth, the effect of climate change and global warming, as well as the increasing frequency and intensity of droughts). These consequences are either short-term catastrophes or long-term changes in the climate system [14,15].

Forecasting drought is essential for water supply management, irrigated agriculture, recreational tourism, environmental monitoring, and the environment's health [16,17]. Fung et al. [18], and Anshuka et al. [19] reviewed various models and techniques that have been applied in previous research to forecast drought. No global approach can outperform all models in all areas of study; thus, it is necessary to evaluate each case separately, assessing the performance of each technique or the combination of different approaches in each research field [20].

Traditional models propose that the relationship between dependent and independent (variables) is linear and can be unsuitable for solving practical application issues [21]. As a result, the complex character and non-linearity of the drought process require algorithms that can simulate non-linear time series data. Therefore, artificial intelligence (AI) algorithms in drought forecasting have received important attention [20]. Moreover, various studies have indicated that AI algorithms outperform traditional methods [22,23]. These AI algorithms are, for example, artificial neural networks (ANNs) [24], support vector machines (SVMs) [25], random forests [26], and the adaptive neuro-fuzzy inference system (ANFIS) [27].

AI has also been used for forecasting droughts by employing a multilayer perceptron network for stochastic synthesis of daily rainfall time series that preserve the Hurst coefficient and the autocovariance structure of all time scales [28]. This statistical property is very important in water management applications because it is related to "the tendency of wet years to cluster into multi-year wet periods or of dry years to cluster into multi-year drought periods" [29].

The hybrid model is developed by combining several different methods; one of these approaches serves as the primary model, while the others serve as pre-or post-processing methods [30]. Various studies have demonstrated that hybrid methods are better than standalone methods. Therefore, several studies recommended using hybrid models to increase forecast precision [22,31,32]. Recently, hybrid prediction models have become more popular than standalone models in hydrology. Figure 1 presents the percentage of studies on hybrid ML models that were used to predict drought indices over the last four years.

Different types of hybrid models have been created and successfully used to increase prediction accuracy [33,34]. Therefore, the hybrid models are classified into three categories:

(1) The pre-processing-based hybrid models (PBH): data pre-processing techniques have an efficient effect on improving the quality of data and selecting the best number of predictors that lead to enhancing the precision of the forecasting model [35]. Numerous studies have been conducted by combining the model with data pre-processing to predict drought, e.g., Zhang et al. [22], Belayneh et al. [23], Mathivha et al. [36], and Djerbouai and Souag-Gamane [37].

(2) The parameter optimisation-based hybrid models (OBH): The primary idea behind the OBH models is to use optimisation algorithms to identify the optimal parameters of the models [34], and combine the models with nature-inspired algorithms for hydrological

drought prediction such as Adnan, et al. [32], Nabipour et al. [20], Banadkooki et al. [38], and Kisi et al. [39].

(3) Hybridisation of components combination-based with preprocessing-based hybrid models (HCPH): Combining two models that have the remarkable performance of prediction models with preprocessing-based hybrid models [34]. Few studies employed this type of hybrid model, such as Khan et al. [31] and Wu et al. [40].

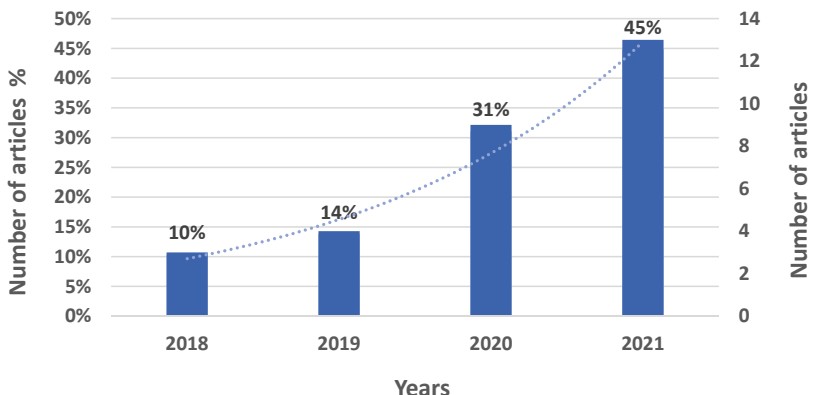

**Figure 1.** A percentage of studies on hybrid ML models used to predict drought indices over the last four years.

Based on our knowledge, three recent review papers have investigated drought forecasting. Anshuka et al. [19] focused on the Standardised Precipitation Index (SPI), the most widely used drought indicator that the World Meteorological Organisation recommends, and a meta-analysis was conducted to test the suitability of data-driven models for predicting SPI. The main goal of the second review paper is to give researchers a quick overview of the models' principles and historical uses. This would keep them from overlooking a possible choice of models and save them a lot of time on the problem [18]. Sundararajan et al. [8] offered a literature survey using machine learning in drought prediction, performance metrics, drought indices, and datasets. Therefore, in addition to the previous literature review, this paper focuses on data pre-processing, data-driven modelling strategies (i.e., univariate or multivariate), machine learning algorithms (i.e., advantages and disadvantages), hybrid models, and performance metrics. The general framework of drought prediction and evaluation is shown in Figure 2.

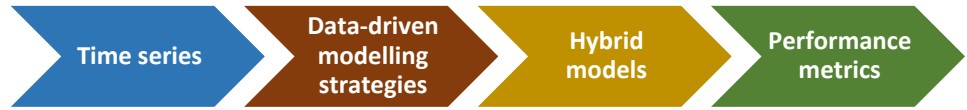

**Figure 2.** The general framework of drought prediction and evaluation.

## 2. Drought

Drought is an unforecastable natural calamity that may cause significant threats such as water shortages and ecosystem deterioration. In addition, water is a vital resource for agriculture and many other human activities. Thus, a water shortage may negatively impact agricultural productivity and threaten food security. Water shortage is a significant concern in different parts of the world [32,41]. Erhardt and Czado [42] mentioned that there are different types of droughts:

- Hydrological (water supplies are dwindling);
- Meteorological (shortage of rainfall);
- Agricultural (soil moisture deficiency);
- Groundwater (decreased levels, discharge, and recharge of groundwater); and
- Socio-economic (an excess demand for commodities due to water scarcity), which is driven by several variables.

The prediction of drought is difficult because of its variability. Thus, it is vital to employ robust and accurate predicting models [38,43]. Drought forecasting is imperative since it allows for early warnings and hence more effective risk management. Different climatic variables are used as inputs to the forecasting model. The technique given here may be beneficial for decision-makers in reducing its consequences and developing effective strategies for water resource management on a monthly or seasonal timeline [32,44]. Agricultural production, the economy, and the environment are all negatively affected by their impacts. For example, from 1999 to 2000, a significant drought in Central and South-West Asia affected 60 million people and resulted in an estimated $4.29 billion in economic losses [21,45].

Drought indicators are a scale of hydrological, meteorological, agricultural, or socio-economic variables that may be used to reference probable its-related stress or deficiency; its' indices attempt to determine the extent of the drought's reach [46,47], and it is considered an effective instruments for making decisions rather than relying on raw data [48]; despite the existence of several indices, there is no widely accepted index for all kinds of droughts that take a wide diversity of drought producing factors into consideration [49,50]. The indices include the Standardized Precipitation Index (SPI) [51], Standardized Precipitation Evapotranspiration Index (SPEI) [5,52], Groundwater Resource Index (GRI) [38], Standard Index of Annual Precipitation (SIAP) [53], Precipitation Index (PI) [13], Standardized Hydrological Drought Index (SHDI) [20], Reconnaissance Drought Index (RDI) [54], Streamflow Drought Index (SDI) [55], and Palmer Drought Severity Index (PDSI) [56]. Tables 1–4 present the classification of SPI, PI, SIAP, and PDSI indices.

**Table 1.** Classification of drought based on SPI [57].

| SPI Values | Class |
| --- | --- |
| >2 | Extremely wet |
| 1.5 to 1.99 | Very wet |
| 1.0 to 1.49 | Moderately wet |
| −0.99 to 0.99 | Near normal |
| −1.0 to −1.49 | Moderately dry |
| −1.5 to −1.99 | Severely dry |
| <−2 | Extremely dry |

**Table 2.** Classification of drought based on PI [13].

| PI Values | Drought Description |
| --- | --- |
| >1.0 | Extremely wet |
| 0.75 to 1.0 | Very wet |
| 0.5 to 0.75 | Moderately wet |
| 0.5 to −0.5 | Normal |
| −0.5 to −0.75 | Moderate drought |
| −0.75 to −1.0 | Severe drought |
| <−1.0 | Extreme drought |

**Table 3.** Classification of drought based on SIAP [53].

| SIAP Values | Classes of Drought Intensity |
| --- | --- |
| >0.84 | Extremely wet |
| 0.52 to 0.84 | Wet |
| −0.52 to 0.52 | Normal |
| −0.52 to −0.84 | Drought |
| <−0.84 | Extreme drought |

**Table 4.** Classification of drought based on PDSI [58].

| PDSI Values | Category |
|---|---|
| PDSI $\leq -4$ | Extreme drought |
| $-4 <$ PDSI $\leq -3$ | Severe drought |
| $-3 <$ PDSI $\leq -2$ | Moderate drought |
| $-2 <$ PDSI $\leq -1$ | Mild drought |
| PDSI $\geq -1$ | No drought |

## 3. Data-Driven Modelling Strategies

Data-driven modelling strategies generally consist of univariate and/or multivariate models [59]. Based on previous studies, Table 5 summarises the comprehensive details of two strategies: univariate (i.e., input-itself-output) and multivariate (i.e., more than one input variable and another output). For instance, a univariate model might just use a drought index (same time series) as input and output, while a multivariate model would utilise several parameters (e.g., rainfall, temperature, evaporation, etc.) as input and a drought index as output. When data needed for forecasting are available, such as precipitation, temperature, evaporation, and other factors influencing drought, these data can be used as input variables. When only drought index data are available, predicting drought index changes must rely only on drought index series with a specific number of time lags [59,60]. The table demonstrates that most previous studies applied a univariate strategy, and a few used a multivariate method.

**Table 5.** Different input strategies.

| Ref. | Type | Input Data Parameter | Output Parameter |
|---|---|---|---|
| [61] | Multivariate | Twelve multivariate datasets (derived from statistically significant lagged combinations of precipitation, temperature, and humidity) | SPI1, SPI 3, SPI 6, SPI 12 |
| [31] | Multivariate | Rainfall data series with SPI Lag | SPI $_{(t+1)}$ |
| [35] | Multivariate | antecedent SPIs and antecedent accumulated monthly rainfall | SPI 3, SPI 6 |
| [20] | Multivariate | SHDI Lag, SPI Lag, and precipitation. | SHDI 1, SHDI 3, SHDI 6. |
| [53] | Multivariate | Rainfall and SIAP Lag<br>Water level and SWSI Lag | SIAP<br>SWSI |
| [62] | Univariate | SPI Lag | SPI 6, SPI 12 |
| [58] | Univariate | PDSI Lag | PDSI $_{(t+1)}$ |
| [63] | Univariate | MSPI Lag | MSPI $_{(t+1)}$ |
| [64] | Univariate | SPI Lag | SPI $_{(t+1)}$ |
| [65] | Univariate | EDI Lag | EDI $_{(t+1)}$ |
| [39] | Univariate | SPI Lag | SPI 3, SPI 6, SPI 9, SPI 12 |
| [66] | Univariate | EDI Lag | EDI $_{(t+1)}$ |
| [67] | Univariate | sc-PDSI Lag | sc-PDSI $_{(t+1),(t+3),(t+6)}$ |
| [38] | Univariate | GRI Lag | GRI 6, GRI 12, GRI 24 |
| [32] | Univariate | SPI Lag | SPI 3, SPI 6, SPI 9, SPI12 |
| [25] | Univariate | PDSI Lag | PDSI $_{(t)}$ |
| [68] | Univariate | SPI Lag | SPI 3 |
| [37] | Univariate | SPI Lag | SPI 3, SPI 6, SPI 12 |
| [54] | Univariate | RDI Lag | RDI 6, RDI 9, RDI 12 |
| [52] | Univariate | SPEI Lag | SPEI 1, SPEI 3, SPEI 6 |
| [27] | Univariate | sc-PDSI Lag | sc-PDSI 1, sc-PDSI 3, sc-PDSI 6 |

## 4. Machine Learning Algorithms

Machine learning (ML) algorithms are an implementation of Artificial Intelligence (AI). ML algorithms are designed with the ability to learn from previous experiences and perform new tasks. These algorithms are typically divided into supervised learning and unsupervised learning. Supervised learning employs labelled training data to produce output data based on past experience. Unsupervised learning is challenging since the system only has unlabeled data, yet it works independently to find the information [8]. Several machine learning algorithms are employed in prediction, such as artificial neural networks (ANNs) [53], adaptive neuro-fuzzy inference system (ANFIS) [39], support vector regression (SVR) [69], and random forest (RF) [61], gradient boosting (GB) [70], k-Nearest Neighbour (k-NN) [71], and decision tree (DT) [72].

Table 6 shows the advantages and disadvantages of the most popular machine learning algorithms.

**Table 6.** Advantages and disadvantages of stand-alone machine learning models.

| Model Type | Advantages | Disadvantages | Ref. |
|---|---|---|---|
| ANN | -Ability to simulate and predict non-stationary and non-linear time series. | -Sometimes, ANNs have issues forecasting unstable and non-stationary time series. If data pre-processing does not apply, the ANN will be unable to forecast and solve issues. | [48,73] |
| ANFIS | -Use the fuzzy logic and neural network in a single model to increase efficiency. | -It needs a lot of training data to create a precise model, and these data may not be available every time. | [27,74] |
| RF | -Accuracy in modelling improves as the number of trees increases. -Ability to process large datasets involving several features | -Applying the model with a large number of trees causes a slow training process. | [26,75] |
| SVR | -It has flexibility for multiple options due to the availability of different kernel functions. | -It needs effective parameter optimisation to provide more accurate predictions. | [75,76] |

## 5. Data Pre-Processing

Despite the high accuracy of different models in estimating hydroclimate parameters (for example, rainfall), these models may have limitations when dealing with hydrological time series that are often non-stationary and cover a wide variety of scales, ranging from a few minutes to several decades. Data pre-processing may be necessary to overcome comparable defects and problems [14,77]. According to Maier and Dandy [78], it is essential to pre-process data suitably before using it in ANN. These strategies are necessary to ensure that all data is treated equally in the learning model. Data pre-processing is vital for the majority of hydrologic time series to attain higher prediction performance [79]. It has been effectively utilised in various fields of study, e.g., monthly rainfall forecasting [80], urban water demand prediction [81], irrigation water prediction [82], and drought forecasting [83]. Data pre-processing includes normalisation, cleaning, and selecting the best model input [84]. It can be divided into three parts.

### 5.1. Normalisation

Screening continuous factors for normality is an essential early stage in the analysis, and the histograms are one of the most important ways of assessing normality [85]. In ML approaches, data curation is a necessary pre-processing step, beginning with data normali-

sation to restrict the input value range [86]. As a result, converting continuous variables is essential for creating time series that are normally or close to normally distributed [87]. Therefore, data normalisation seeks to obtain the identical range of values for each model's inputs and produces a time series normally or nearly normally distributed [88]. The natural logarithm was utilised to normalise the data to decrease the impact of multicollinearity between input variables [89]. In addition, Z-score normalisation is a traditional standardisation approach that uses the mean (μ) and standard deviation (σ) to standardise parameters [90]; it is calculated using Equation (1).

$$Z - score = \frac{Xi - \mu}{\sigma} \tag{1}$$

where *x* is the actual value of the dataset parameter *i*.

Moreover, Min-Max normalisation (*Xnorm*) is one of the most common and widely used data normalisation methods. It is a method that linearly transforms variables, where min and max represent the minimum and maximum values of *x* [91], as shown in Equation (2).

$$Xnorm = \frac{x - \min(x)}{\max(x) - \min(x)} \tag{2}$$

Moreover, Decimal scaling (*Xi'*) is moving the decimal point of the variable's values to achieve the normalised value. The number of decimal points moved depends on the maximum absolute of value of the variables [92]; it is computed by the Equation (3).

$$Xi' = \frac{Xi}{10^k} \tag{3}$$

*k* is the smallest integer such that Max ($\lfloor x \rfloor$) < 1.

### 5.2. Cleaning

Data cleaning techniques include treating the outliers and pre-treatment signals [87]. An outlier is a case in which one variable has an extreme value that leads to distorted statistics. The box and whisker method was employed to identify outliers, which were then treated. This method significantly improves the accuracy of the suggested prediction model [93]. Additionally, there are various noise components in each time series, and the pre-treatment signal approaches (i.e., Wavelet [94], Empirical Mode Decomposition [95], Singular Spectrum Analysis [96], etc.) are the most effective methods to denoise the original time series by analysing them into multiple components [88]. As shown in Equation (4), A time series (T) could be analysed into a trend (X), stochastic (Y), seasonal (Z), and noise (N) [30].

$$T = X + Y + Z + N \tag{4}$$

### 5.3. A Selecting Appropriate Descriptors

Selecting appropriate predictors to simulate hydrological variables is difficult, specifically in non-linear hydrologic systems [68,97]. In general, choosing the optimal model input is one of the most significant steps in data pre-processing and developing an acceptable forecast model [87]. When the data-driven modelling strategy is univariate, different methods are used to choose the best antecedent lags scenario, e.g., the Mutual Information technique [88]. On the other hand, various approaches are employed when the strategy is multivariate, such as dimensionality reduction, which can be achieved by retaining input features that have large variances and discarding those terms that have small variances (i.e., Principal Component Analysis (PCA) is a feature selection approach that can reduce the model's dimensionality without impacting its performance.) [86]. Additionally, variance inflation factor and tolerance approaches are used to determine potential multicollinearity and exclude inputs from ML algorithms.) [14,93,97]. Moreover, automated weighting techniques can be used, such as Bayesian and frequentist algorithms [98]. According to

Tabachnick and Fidell [85], the required sample size for the prediction model is based on the number of predictors (independent variables), as shown in Equation (5) (e.g., if the number of predictors is four variables, the model needs 104 + 4 = 108 cases).

$$N \geq 104 + V \tag{5}$$

where: N: sample size.

V: number of predictors.

Based on previous studies reviewed in this research, Table 7 summarises data pre-processing. Most of these studies used one or two data pre-processing steps; furthermore, several studies did not employ the best model input. Besides, a few studies employed all the data pre-processing steps.

**Table 7.** Summary of data preprocessing.

| Authors | Normalisation | Cleaning | Best Model Input |
|---|---|---|---|
| Danandeh Mehr et al. [99] | Yes | No | Yes |
| Ali et al. [61] | Yes | Yes | Yes |
| Aghelpour et al. [58] | Yes | No | No |
| Aghelpour et al. [63] | Yes | No | No |
| Safavi et al. [64] | Yes | Yes | No |
| Fung et al. [100] | Yes | Yes | No |
| Zhang et al. [101] | No | Yes | Yes |
| Khan et al. [31] | No | Yes | No |
| Pham et al. [35] | Yes | Yes | Yes |
| Banadkooki et al. [38] | Yes | No | Yes |
| Adnan et al. [32] | Yes | No | No |
| Nabipour et al. [20] | Yes | No | No |
| Mohamadi et al. [68] | Yes | No | Yes |
| Djerbouai and Souag-Gamane [37] | Yes | Yes | No |
| Wu et al. [40] | Yes | Yes | No |
| Soh et al. [52] | Yes | Yes | No |
| Başakın et al. [27] | Yes | Yes | No |
| Das et al. [62] | Yes | Yes | No |
| Belayneh et al. [23] | Yes | Yes | No |
| Kisi et al. [39] | Yes | No | Yes |

## 6. Hybrid Models

Hybrid models are a relatively new class of hydrological models that have grown in popularity in recent decades. Fung et al. [18] mentioned the first drought forecasting hybrid model in the hydrology field in 2007. One or more strategies are combined to form a hybrid model; one of these techniques works as the main model, while the others are used as pre-processing or post-processing approaches [88]. Employing hybrid models in drought prediction can be divided into three categories, namely: pre-processing-based hybrid models (PBH), parameter optimisation-based hybrid models (OBH), and hybridisation of components combination-based with pre-processing-based hybrid models (HCPH) [34]. Different studies applied the hybrid models, as shown in Figure 3 and Table 8.

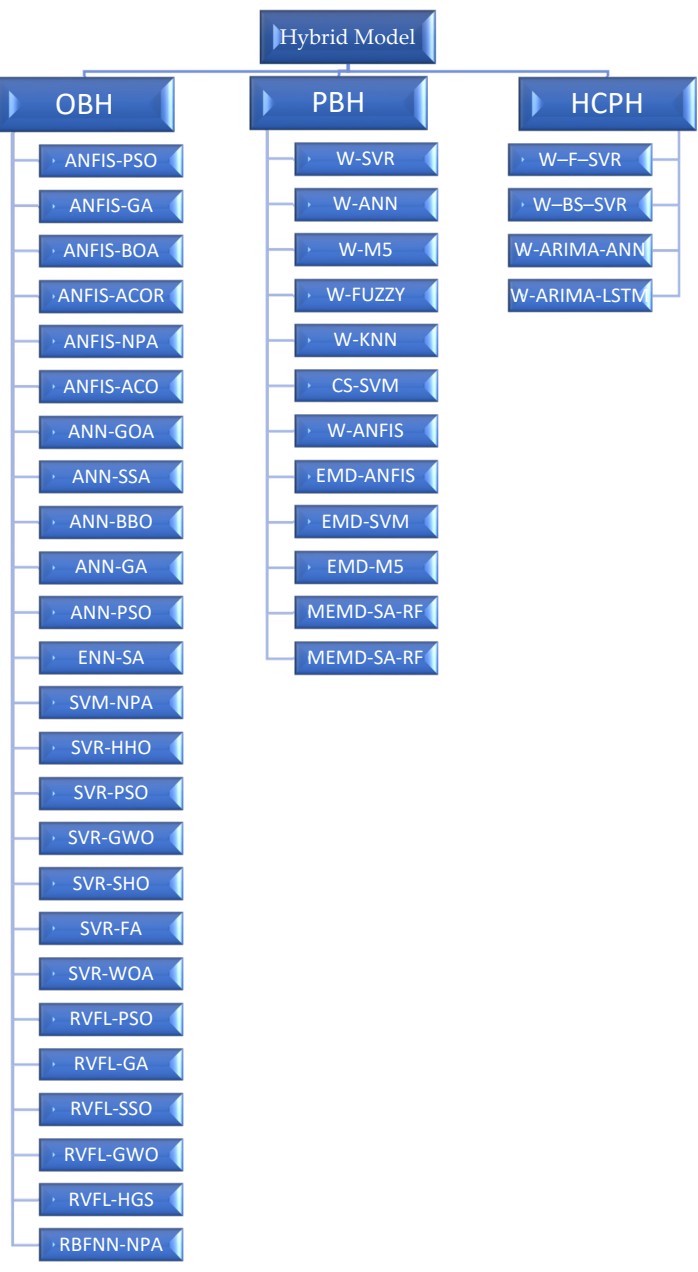

**Figure 3.** Classification of hybrid models that were employed for forecasting drought indices.

*6.1. Pre-Processing-Based Hybrid Models (PBH)*

Data pre-processing is important for the majority of hydrologic time series to attain higher prediction performance [79]. In pre-processing-based hybrid models, the input data is pre-processed by different methods, as mentioned in Section 5.

A drought forecasting model that contains singular spectrum analysis (SSA) and a single least square support vector machine (LSSVM) model. Monthly Standardised Precipitation Index (SPI) data for the Tseng-Wen reservoir catchment in southern Taiwan (1975–2015). The findings indicate that SSA-LSSVM2 (LSSVM-based model combining with SSA utilising the antecedent SPI as input) outperforms LSSVM1 (LSSVM-based approach using antecedent SPI as an input). Additionally, when the performance of SSA-LSSVM2 and SSA-LSSVM3 (SSA-LSSVM-based approach using antecedent cumulative monthly rainfall as an input) is compared, it is determined that SSA-LSSVM3 is the best-suited model for SPI drought forecasting [35].

Two model types, namely: Artificial Neural Network (ANN) and Support Vector Regression (SVR). Moreover, the Wavelet Packet Transform (WPT) approach was used for pre-processing data. The Standardised Precipitation Index (SPI) from 1985 to 2013 in India was used to build and assess the model. The findings indicate that hybrid strategies (i.e., WPT-ANN and WPT-SVR) outperform single machine learning approaches (i.e., ANN and SVR). For the majority of grid sites, the hybrid WPT-ANN model outperformed the WPT-SVR model [62].

An artificial neural network (ANN) model forecasts hydrological and meteorological drought indices. These indices contain the Standardised Water Storage Index (SWSI) and the Standard Index of Annual Precipitation (SIAP). Additionally, the study applied the wavelet (W) for denoising the raw data. The data of the Langat River Basin, which is situated in central Malaysia, over 30 years (1986–2016) was employed to conduct the study. The results demonstrate that the W method increases the quality of raw data. Moreover, the combined model (W-ANN) can simulate the SWSI and SIAP indices with R equal to 0.940 and 0.973, respectively [53].

The Discrete Wavelet Transform (DWT) tool combines separately with k-Nearest Neighbour (kNN), Fuzzy and Support Vector Machine (SVM) approach tools to increase the accuracy of drought predictions using the Palmer drought severity index (PDSI). The models employ data from Turkey's Marmara area from 1960 to 2016. This research demonstrates that combined, WkNN, W-Fuzzy, and W-SVM models outperform stand-alone models (i.e., KNN, Fuzzy, and SVM models) in prediction performance. The prediction performance of the W-Fuzzy model is a little better than that of the W-KNN and W-SVM models [25].

A hybrid wavelet-neural network (W-ANN) model to simulate the standard precipitation index (SPI 6, SPI 12) in the north of the Haihe River Basin, China, from 1960 to 2010. The hybrid model was compared with the ANN and ARIMA methods. The outcomes reveal that W-ANN is the best suited based on several performance metrics [22].

The combined Wavelet (W) with neural networks (W-ANN) technique to forecast the standard precipitation index (SPI) at three-time scales (SPI 3, SPI 6, and SPI 12) in Algeria's Algerois basin (1936–2008). The combined model was compared with the standalone, ANN, and two traditional stochastic models (ARIMA and SARIMA). The results indicated that the wavelet method improves data quality and W-ANN offers the best performance based on different statistical criteria (i.e., NSE, RMSE, and MAE) [37].

The two machine learning models are support vector regression (SVR) and artificial neural networks (ANNs) with and without using wavelet transformations (WT) to predict the Standard Precipitation Index (SPI) for the long term. Moreover, these techniques were compared with the traditional model (ARIMA). The data for Awash River Basin, Ethiopia, from 1970 to 2005, were employed to build and assess the models. The statistical indicators, RMSE, MAE, and $R^2$, demonstrate that the WT-ANN technique outperformed all other models in predicting SPI 12 and SPI 24 [102].

The wavelet transform approach was adopted to denoise the raw data. Two machine learning models (ANN and SVR) and one conventional model (ARIMA) were used to forecast the drought indices. The data for Awash River Basin, Ethiopia, from 1970 to 2005, were employed to simulate short-term drought indices (SPI 3 and SPI 6). Several performance indicators (RMSE, MAE, and $R^2$) were utilised to compare the predicted outcomes of the data-driven models. The WT method demonstrate a capability to clean raw data from noise and the hybrid models (WT-ANN and WT-SVR) outperformed standalone models (ANN, SVR, and ARIMA). Additionally, WT-ANN delivers the best performance for forecast drought [23].

An adaptive neuro-fuzzy inference system (ANFIS) with and without empirical mode decomposition (EMD) to forecast the self-calibrated Palmer Drought Severity Index (sc-PDSI). EMD analyses time series into sub-series and applies ANFIS to each sub-series to forecast the drought index from 1900 to 2016 for Adana, Turkey. The statistical indicators,

NSE and MSE, demonstrate that the hybrid EMD-ANFIS model is better than the ANFIS model when used alone [27].

Three models were used, namely: support vector regression (SVR), artificial neural network (ANN), and hybrid Wavelet and artificial neural network (W-ANN), to predict SPI 3 and SPI 12 in Ethiopia's Awash River Basin from 1970 to 2005. All models' performances were assessed using MAE, RMSE, and $R^2$. The results demonstrate that wavelet neural networks (W-ANN) are the best technique for drought prediction [24].

### 6.2. Parameter Optimisation-Based Hybrid Models (OBH)

New evolutionary algorithms were used to speed up the convergence of the soft computing models and increase their accuracy [68]. In hydrological research, bio-inspired optimisation approaches have been effectively used to improve models' abilities [54]. The parameters of ML models were determined by using optimisation algorithms [38]. Various types of metaheuristic algorithms are used to solve such optimisation problems. The metaheuristic algorithms emulated mathematical methods by following natural phenomena, such as a physical annealing process, animal behaviour, biological evolutionary process, etc. [103].

The ANN models with three meta-heuristic optimisation algorithms, namely the salp swarm algorithm (SSA), particle swarm optimisation (PSO), and genetic algorithm (GA), to predict the groundwater resource index (GRI). The study area is the Yazd plain, Iran, at different timescales (6, 12, and 24 months) (180 months). The GRI was modelled using five input scenarios: GRI-1 lagged, GRI-2 lagged, GRI-3 lagged, GRI-4 lagged, and GRI-5 month-lagged. The result revealed that hybrid models PSO-ANN (5), SSA-ANN (5), and GA-ANN (5) exceed other hybrid ANN models. Furthermore, a lower mean absolute error (MAE) was found in ANN-SSA models using an input scenario (5) [38].

The random vector functional link (RVFL) combined with the particle swarm optimisation (PSO), genetic algorithm (GA), grey wolf optimisation (GWO), social spider optimisation (SSO), salp swarm algorithm (SSA), and hunger games search algorithm (HGS) to predict SPI (SPI_3, SPI_6, SPI_9, and SPI_12). The data was collected from three stations in Bangladesh for 30 years. The results reveal that the HGS algorithm performed better than the other algorithms and significantly increased the RVFL method's accuracy in drought forecasting [32].

A hybridisation of artificial neural networks (ANN) with novel nature-inspired optimisation algorithms, includes the grasshopper optimization algorithm (GOA), salp swarm algorithm (SSA), biogeography-based optimisation (BBO), and particle swarm optimization (PSO) for simulating short-term hydrological drought, from October 1963 to September 2017. The Standardised Hydrological Drought Index (SHDI) was utilised to predict drought under scenarios of SHDI 1, SHDI 3, and SHDI 6 in the Dez dam in the southwestern region of Iran. According to the findings, the hybridised model outperformed the traditional ANN. Overall; PSO outperformed the other optimisation techniques in terms of performance [20].

The adaptive neuro-fuzzy inference system (ANFIS) combined with different meta-heuristic algorithms, including the genetic algorithm (GA), ant colony algorithm (ACO), particle swarm optimisation (PSO), and butterfly optimisation algorithm (BOA) to forecast different time scales of SPI (SPI_3, SPI_6, SPI_9, and SPI_12). Monthly precipitation data from 1985 to 2015 in Iran were used in this study. The results demonstrate that hybrid models are better than the standalone model. In addition, the ANFIS-PSO method achieves the highest degree of accuracy [39].

Four types of machine learning methods were employed to predict SPI 3 in Iran, using data from 1980 to 2014. These methods are the adaptive neuro-fuzzy interface system (ANFIS), radial basis function neural network (RBFNN), multilayer perceptron (MLP), and support vector machine (SVM). The above methods were hybridised with different metaheuristic algorithms, including a nomadic people algorithm (NPA), the slap swarm algorithm (SSA), the bat algorithm (BA), and the krill algorithm. Generally, single SVM,

RBFNN, ANFIS, and MLP models performed less well than hybrid models. Among all the hybrid models, the ANFIS–NPA method performed the best based on statistical criteria [68].

*6.3. Hybridisation of Components Combination-Based with Preprocessing-Based Hybrid Models (HCPH)*

The components combination-based hybrid models (CBH) models aim to improve the prediction accuracy by combining the remarkable abilities of single predicting models [33]. The classic CBH models typically consist of linear and nonlinear components and are simulated by statistical and intelligent models [104]. CBH models are combined with preprocessing-based hybrid models (PBH) to create HCPH models to discard indeterminacy and extract useful information from raw data [34].

There are three scenarios to use ANN for predicting drought. The first one uses ANN as a standalone model. Second a combined model includes WT-ANN. Finally, a novel technique contains WT-ARIMA-ANN. This study analysed 30-year rainfall data for Malaysia's Langat River Basin from 1986 to 2016. The WT-ARIMA-ANN technique offers the best scenario to forecast drought [31].

The model wavelet-ARIMA long-short-term memory (WT-ARIMA-LSTM) was used to estimate precipitation and predicate drought events through the China Z-Index (CZI) based on annual precipitation. The data used are monthly from 1967 to 2017 for three stations in China. The hybrid technique (WT-ARIMA-LSTM) was compared separately with the standalone models ARIMA and LSTM. The results presented that the hybrid model had greater prediction accuracy than the ARIMA and LSTM at varying training and test sets [40].

The WT method to clean raw data and ANFIS and hybrid ARIMA-ANN models were applied to forecast SPEI (SPEI 1, SPEI 3, and SPEI 6) in the Langat River Basin in Malaysia from 1976 to 2015. The models were evaluated using RMSE, MAE, $R^2_{adj}$, Nash-Sutcliffe Coefficient of Efficiency (NSE), and Willmott's Index of Agreement. The hybrid technique (WT-ARIMA-ANN) forecasts the drought index better for both the short and mid-term [52].

The following is a summary of findings from a review of several articles:

1. Numerous studies have demonstrated that combined metaheuristics with ML models outperform single-model approaches.
2. Combining decomposition techniques with machine learning (ML) models may be used to increase the performance of ML models.
3. The wavelet method has been proven to be successful in denoising raw data, increasing the results' accuracy.

**Table 8.** Summary of studies applying machine learning models for forecasting drought indices.

| Authors | Region | Size of Data | Model | Best Model | Performance Metric |
|---|---|---|---|---|---|
| Malik et al. [66] | India | 1901–2015 | SVR-HHO, SVR-PSO | SVR-HHO | RMSE, MAE, COC, NSE, WI |
| Taylan et al. [105] | Turkey | 1975–2010 | ANFIS, SVM, ANN, W-ANFIS, W-SVM, W-ANN | W-ANFIS | $R^2$, RMSE, K–S test |
| Adnan et al. [32] | Bangladesh | 30 years | RVFL, RVFL- (PSO, GA, GWO, SSO, SSA, HGS) | RVFL-HGS | RMSE, MAE, NSE, $R^2$ |
| Aghelpour et al. [58] | Iran | 1960–2018 | SVM-DA, ARMA, RBFNN, SVM | SVM-DA | RMSE, NRMSE, WI, R, MAE |
| Pham et al. [35] | Taiwan | 1975–2015 | LSSVM1, SSA-LSSVM2, SSA-LSSVM3 | SSA-LSSVM3 | RMSE, MAE, R |
| Ahmadi et al. [54] | Iran | 1974–2018 | SVR, SVR-FA, SVR-WOA, W-SVR | W-SVR | RMSE, MAE, WI, NSE |

**Table 8.** *Cont.*

| Authors | Region | Size of Data | Model | Best Model | Performance Metric |
|---|---|---|---|---|---|
| Altunkaynak and Jalilzadnezamabad [25] | Turkey | 1960–2016 | Fuzzy, kNN SVM, W-Fuzzy, W-kNN, W-SVM | W-Fuzzy | MSE, CE, $R^2$ |
| Wu et al. [40] | China | 1967–2017 | wavelet-ARIMA-LSTM, ARIMA, LSTM | wavelet-ARIMA-LSTM | RMSE, MAE, $R^2$ |
| Malik et al. [65] | India | 1901–2015 | SVR–GWO, SVR–SHO | SVR–GWO | MAE, RMSE, NSE, WI, R |
| Danandeh Mehr et al. [106] | Turkey | 1971–2016 | WPGP, AR1, GP, RF | WPGP | NSE, RMSE |
| Banadkooki et al. [38] | Iran | 15 years | ANN–SSA, ANN–PSO, ANN-GA | ANN–SSA | NSE, RMSE, MAE |
| Xu et al. [107] | China | 1951–2017 | ARIMA, SVR, LSTM, ARIMA-SVR, LS-SVR, ARIMA-LSTM | ARIMA-LSTM | NSE, MSE, MAE, RMSE |
| Alquraish et al. [108] | Saudi Arabia | 1968- 2019 | ARIMA, ARIMA–GA, HMM HMM-GA, ARIMA–GA–ANN | ARIMA–GA–ANN | RMSE, $R^2$, NSE, MAD |
| Nabipour et al. [20] | Iran | 1963–2017 | ANN-GOA, ANN-SSA, ANN-BBO, ANN-PSO, ANN | ANN-PSO | RMSE, $R^2$ |
| Das et al. [62] | India | 1985–2013 | ANN, SVR, WPT -ANN, WPT-SVR | WPT -ANN | $R^2$, RMSE, MAE |
| Xu et al. [109] | China | 1951–2017 | ARIMA, ARIMA–SVR | ARIMA–SVR | RMSE, MAE, $R^2$, NSE |
| Danandeh Mehr et al. [99] | Turkey | 1971–2016 | ENN, ENN-SA | ENN-SA | NSE, MXL, RMSE, BIC |
| Khan et al. [31] | Malaysia | 1986–2016 | ANN, wavelet-ANN, Wavelet-ARIMA-ANN | Wavelet-ARIMA-ANN | R, RMSE, $R^2$ |
| Mohamadi et al. [68] | Iran | 1980–2014 | ANFIS, ANFIS-NPA, MLP–NPA, RBFNN-NPA, SVM–NPA | ANFIS-NPA | NSE, RMSE, MAE, PBIAS, $R^2$ |
| Özger et al. [67] | Turkey | 116 years | M5, ANFIS, SVM, W-ANFIS, W-SVM, W-M5, EMD-ANFIS, EMD-SVM, EMD-M5 | W-ANFIS | MSE, NSE, $R^2$ |
| Aghelpour et al. [63] | Iran | 59 years | ANFIS, ANFIS-ACO, ANFIS-GA, ANFIS-PSO | ANFIS-ACO | RMSE, MAE, WI |
| Başakın et al. [27] | Turkey | 1900–2016 | ANFIS, EMD-ANFIS | EMD-ANFIS | MSE, NSE |
| Fung et al. [100] | Malaysia | 1976–2015 | W–BS–SVR, multi-input-W–F–SVR, weighted-W-F–SVR | Weighted-W–F–SVR | RMSE, $R^2$, MAE |
| Kisi et al. [39] | Iran | 1985–2015 | ANFIS, ANFIS-PSO, ANFIS-GA, ANFIS-BOA, ANFIS-ACOR | ANFIS-PSO | RMSE, MAE, IA |
| Ali et al. [61] | Pakistan | 1981–2015 | MEMD-SA-RF, KRR, RF MEMD-SA-KRR | MEMD-SA-RF | MSE, R, RMSE |

**Table 8.** *Cont.*

| Authors | Region | Size of Data | Model | Best Model | Performance Metric |
|---|---|---|---|---|---|
| Zhang et al. [101] | China | 1979–2016 | ARIMA, W-ANN, SVM | ARIMA | $R^2$, MSE, NSE, K–S |
| Khan et al. [53] | Malaysia | 1986–2016 | ANN, W-ANN | W-ANN | R, RMSE |
| Safavi et al. [64] | Iran | 1969–2009 | W-SVM, CS-SVM, SVM | W-SVM | $R^2$, RMSE |
| Soh et al. [52] | Malaysia | 1976–2015 | Wavelet-ARIMA-ANN, W-ANFIS | Wavelet-ARIMA-ANN | $R^2_{adj}$, RMSE, MAE, NSE |
| Zhang et al. [22] | China | 1960–2010 | ARIMA, ANN, W-ANN | W-ANN | K–S, $R^2$, Kendall rank correlation |
| Djerbouai and Souag-Gamane [37] | Algeria | 1936–2008 | ANN, W-ANN, ARIMA, SARIMA | W-ANN | NSE, RMSE, MAE |
| Deo et al. [110] | Australia | 1916–2012 | ELM, ANN, LSSVR, W-ANN, W-LSSVR, W-ELM | W-ELM | $R^2$, WI, NSE, RMSE, MAE, $P_{dv}$ |
| Belayneh et al. [23] | Ethiopia | 1970–2005 | ARIMA, ANN, SVR, W-SVR, W-ANN | W-ANN | RMSE, MAE, $R^2$ |

## 7. Performance Metrics

In a drought forecast, it is necessary to evaluate the overall performance and capacity of the prediction model. The criteria used to assess the efficacy of the forecasting model are essential since it impacts the decision to select the best model or scenario [111]. The following are the most applied criteria used in the previous studies:

Root mean square error (RMSE) [112];
Mean absolute error (MAE) [113];
Determination coefficient ($R^2$) [114];
The correlation coefficient (R) [115];
Nash-Sutcliffe-efficiency (NSE) [116];
Mean percentage error (MPE) [117];
Scatter index (SI) [118];
Bayesian information criterion (BIC) and Akaike information criterion (AIC) [119];
Absolute average deviation (AAD) [120].

### 7.1. Mean Absolute Error

The mean absolute error (MAE) is a statistical term that refers to the total of the absolute variations between predicted and measured [113]. The formula for calculating MAE is in the following Equation (6):

$$MAE = \frac{1}{N} \sum_{i=1}^{N} | Ri - Pi| \tag{6}$$

where: *Pi*: Represents predicted value;
*Ri*: Represent real value;
$\overline{R}_i$: Represent mean of real value;
$\overline{P}_i$: Represent mean of predicted value;
*N*: Represents the total number;
*i*: Represents a single data.

### 7.2. Root Mean Squared Error

The root mean squared error (RMSE) is the average squared variation between the predicted and actual output. It finds the data concentration around the best fit line [112,121].

It is employed whenever the error is significantly non-linear; this is a good indicator of the accuracy of a forecast [8].

$$RMSE = \sqrt{\sum_{i}^{N} \frac{(Ri - Pi)^2}{N}} \tag{7}$$

### 7.3. Determination Coefficient

The training-validation data set and the testing data set are being used. The coefficient of determination ($R^2$) is a statistical measurement of the correlation between the observed and expected values in a given situation. Using $R^2$ values, which range between 0 and 1, the complete relationship between the data set and the line drawn across them is shown by 1, and no relation to the type between the data and the line drawn through them is indicated by 0 [114,122].

$$R^2 = \left( \frac{\sum_{i=1}^{N}(Ri - \overline{Ri})(Pi - \overline{Pi})}{\sqrt{\sum(Ri - \overline{Ri})^2 \sum(Pi - \overline{Pi})^2}} \right)^2 \tag{8}$$

### 7.4. Nash-Sutcliffe Efficiency

In 1970, Nash and Sutcliffe designed the Nash-Sutcliffe efficiency (NSE). Additionally, it evaluates the accuracy of hydrological models [8,116]. The NSE is sensitive to variations between forecasts and observations that are both additive and proportional in magnitude. As a result, since NSE squares the values of paired differences, it has a disproportionate sensitivity to extreme values [37].

$$NSE = 1 - \left\{ \frac{\sum_{i=1}^{N}(Ri - Pi)^2}{\sum_{i=1}^{N}(Ri - \overline{Ri})^2} \right\} \tag{9}$$

### 7.5. Mean Percentage Error (MPE)

The mean percentage error (MPE) is the percentage deviation between the measured and predicted [117], as shown in Equation (10).

$$MPE = \frac{1}{N} \sum_{i=1}^{N} \frac{Pi - Ri}{Ri} \tag{10}$$

### 7.6. Scatter Index (SI)

The Scatter Index (*SI*) is dimensionless and an indicator of a model's overall relative accuracy. The model accuracy was considered poor if $SI \geq 30\%$, fair if $20\% \leq SI < 30\%$, good if $10\% \leq SI < 20\%$, and excellent if $SI < 10\%$ [117,118].

$$SI = \frac{RMSE}{\overline{Ri}} \times 100 \tag{11}$$

### 7.7. Bayesian Information Criterion (BIC) and Akaike Information Criterion (AIC)

BIC and AIC are select model metrics, in which a conventional assessment measure is modified based on the number of data points used for calibration, m, and the number of free parameters in each model, p [119]. Equations (12) and (13) are employed to calculate the Bayesian information criterion (BIC) and the Akaike information criterion (AIC).

$$BIC = m.\ln(RMSE) + p.\ln(m) \tag{12}$$

$$AIC = m.\ln(RMSE) + 2p \tag{13}$$

*7.8. Absolute Average Deviation (AAD)*

AAD is the statistical metric that measures the deviation between the model predictions and the experimental results. Generally, the AAD between forecasted and observed findings is preferred to be as small as possible [123].

$$ADD(\%) = \left\{ \frac{1}{N} \sum_{i=1}^{N} \frac{Ri - Pi}{Ri} \right\} \times 100 \tag{14}$$

## 8. Future Research

Banadkooki et al. [38] proposed that future research may consider hybrid ANN models and the influence of climate change on drought modelling. Moreover, various pre-processing strategies for identifying the optimal input scenario for the models might be considered. Adnan et al. [32] applied RVFL, RVFL-PSO, RVFL-GA, RVFL-GWO, RVFL-SSO, RVFL-SSA, and RVFL-HGS to forecast SPI. The recommendation is to make a comparison between the algorithms and models under investigation and other metaheuristic algorithms and/or models. Aghelpour et al. [63] recommended that future studies employ optimisation algorithms such as Firefly, Gray Wolf, Krill herd, etc., to improve ANFIS's ability to forecast MSPI. Pham et al. [35] proposed that oceanic/atmospheric circulation parameters may be used as model inputs for the SSA-LSSVM method in addition to historical precipitation. Moreover, the influence of other data pre-processing techniques (e.g., discrete wavelet transform) on the LSSVM's efficiency might be investigated further. Ahmadi et al. [54] evaluated a single SVR along with the hybrid SVR-WOA, SVR-FA, and W-SVR to predicate the reconnaissance drought index (RDI). The recommendation is that different AI models may be combined with various bio-inspired optimisation techniques and wavelet analysis to develop other kinds of hybrid models.

Moreover, based on analysis of the previous studies in this field, hybrid models can be improved by the following:

1. Using various data pre-treatment techniques, such as singular spectrum analysis (SSA) and empirical mode decomposition (EMD).
2. It is suggested to employ a multivariate strategy.
3. The selection of input variables is critical and influences the performance and accuracy of a model's output. As a result, it is recommended that more efforts should be put into determining the optimal input variable combination scenario. Hence, it is also recommended that other methods should be used to determine the inputs, such as feature selection methods, feature extraction methods, and dimensionality reduction methods.
4. The use of hybrid metaheuristic algorithms and machine learning techniques in drought predicting has grown considerably in recent years. Nevertheless, there is still room for enhancement concerning drought prediction.
5. Applying the hybridisation of pre-processing-based with parameter optimisation-based hybrid models (i.e., including both PBH and OBH).

## 9. Conclusions

This research has reviewed the literature on usually applied data-driven strategies concerning forecast performance and precision. The analysis demonstrated that the performance of various machine learning, probabilistic, and time-series models is typically consistent and comparable.

Various factors affect the prediction technique's performance and accuracy; choice and comparison of techniques were conducted based on data pre-processing, an appropriate timescale, data-driven modelling strategies (univariate or multivariate), and a metaheuristic algorithm combined with the model. Recent research proved that separate classical approaches no longer provide the most accurate findings. Accordingly, hybrid models are the most effective tools that must be used to increase the accuracy of drought predictions. A comprehensive hybrid model incorporates both pre-processing and metaheuristic algorithm

techniques. Thus, a key strength of the present study was it represents a comprehensive examination of all the above factors.

Most of the data on drought is used to develop non-linear predictive models. For this type of data, models that incorporate only proven effective variables are more accurate than models that incorporate all available data without testing variables' efficiency. Consequently, in future research, the efficiency of the variables should be tested (best model input) before using all of the data as an input to the prediction models and applying normalisation and cleaning. Moreover, although significant advances in hybrid model techniques have been made recently, no new techniques, among others, have emerged as the best predicting method. Therefore, drought prediction remains a research problem, which leaves room for researchers to enhance hybrid techniques for specific applications.

**Author Contributions:** Conceptualization, M.A.A. and S.L.Z.; methodology, M.A.A., S.L.Z. and N.A.-A.; investigation, M.A.A.; resources, N.A.-A. and K.H.; writing—original draft preparation, M.A.A.; writing—review and editing, M.A.A., S.L.Z. and N.S.S.A.-B.; visualization, N.S.S.A.-B. and K.H.; supervision, S.L.Z.; project administration, S.L.Z.; funding acquisition, N.A.-A. All authors have read and agreed to the published version of the manuscript.

**Funding:** The APC was funded by Lulea University of Technology.

**Conflicts of Interest:** The authors declare no conflict of interest.

## Abbreviations

| | |
|---|---|
| AAD | Absolute Average Deviation |
| ACO | Ant Colony Optimization |
| AIC | Akaike Information Criterion |
| ANFIS | Adaptive Neuro-Fuzzy Inference System |
| ANN | Artificial neural network |
| AR1 | Autoregressive |
| ARIMA | Autoregressive integrated moving average |
| ARMA | Autoregressive Moving Average |
| ARIMA–GA | Auto-regressive integrated moving average–genetic algorithm |
| BBO | Biogeography-Based Optimisation. |
| BIC | Bayesian Information Criterion |
| BOA | Butterfly Optimization Algorithm |
| BS | Boosting |
| CANFIS | Co-active neuro fuzzy inference system |
| CE | Coefficient of Efficiency |
| CS | Cuckoo Search |
| DA | Dragonfly Algorithm |
| EDI | Effective Drought Index |
| ELM | Extreme learning machine |
| EMD | Empirical Mode Decomposition |
| ENN | Elman Neural Network |
| F | Fuzzy |
| FA | Firefly Algorithm |
| GA | Genetic Algorithm |
| GOA | Grasshopper Optimisation Algorithm. |
| GP | Genetic programming |
| GRI | Groundwater Resource Index |
| GWO | Grey Wolf Optimizer |
| HGS | Hunger Games Search algorithm |
| HHO | Harris Hawks Optimization |
| HMM–GA | Hidden Markov model–genetic algorithm |
| IA | Index of Agreement |
| KA | Krill Algorithm |
| kNN | k- Nearest Neighbour |
| KRR | Kernel Ridge Regression |

| | |
|---|---|
| K–S | Kolmogorov–Smirnov |
| LSSVM | Least Square Support Vector Machine |
| LSSVM1 | LSSVM based model using antecedent SPI as input |
| LSSVR | Least squares support vector regression |
| LSTM | Long Short-Term Memory |
| LS-SVR | Least square- Support Vector Regression |
| M5 | Model Tree |
| MAE | Mean Absolute Error |
| MAD | Mean absolute deviation |
| MEMD | Multivariate Empirical Mode Decomposition |
| MLP | Multilayer Perceptron |
| MLR | Multiple linear regression |
| MPE | Mean Percentage Error |
| MSPI | Multivariate Standardized Precipitation Index |
| MXL | Maximum Likelihood |
| NPA | Nomadic People Algorithm |
| NRMSE | Normalized Root Mean Squared Error |
| NSE | Nash-Sutcliffe coefficient of efficiency |
| Pdv | Percentage peak deviation |
| PBIAS | Percent Bias |
| PDSI | Palmer Drought Severity Index |
| PSO | Particle Swarm Optimisation |
| R | Correlation Coefficient |
| $R^2$ | Coefficient of Determination |
| $R^2_{adj}$ | Adjusted Coefficient of Determination |
| RBFNN | Radial Basis Function Neural Network |
| RDI | Reconnaissance drought index |
| RF | Random Forest |
| RMSE | Root Mean Square Error |
| RVFL | Random Vector Functional Link |
| SA | Simulated Annealing optimization algorithm. |
| SARIMA | Seasonal Autoregressive Integrated Moving Average |
| sc-PDSI | Self-calibrated Palmer Drought Severity Index |
| SHDI | Standardised Hydrological Drought Index. |
| SHO | Spotted Hyena Optimizer |
| SI | Scatter Index |
| SIAP | Standard Index of Annual Precipitation |
| SPEI | Standardized Precipitation Evapotranspiration Index |
| SPI | Standardized Precipitation Index |
| SSA | Singular spectrum analysis |
| SSA | Salp swarm algorithm |
| SSA-LSSVM2 | The LSSVM-based model coupling with SSA using antecedent SPI as input. |
| SSA-LSSVM3 | The SSA-LSSVM-based model using antecedent accumulated monthly rainfall as input was developed and compared to SSALSSVM2. |
| SSO | Social Spider Optimization |
| SVM | Support Vector Machine |
| SVR | Support Vector Regression |
| SWSI | Standardized Water Storage Index |
| W | Wavelet |
| WANN | Wavelet Artificial Neural Network |
| W–BS–SVR | Wavelet–Boosting–Support Vector Regression |
| W–F–SVR | Wavelet–Fuzzy–Support Vector Regression |
| WI | Willmott's Index |
| WOA | Whale Optimization Algorithm |
| WPGP | Wavelet packet-genetic programming |
| WPT | Wavelet Packet Transform |
| WSVM | Wavelet-Support Vector Machine |

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
