# Peer review of "Drought Forecasting: A Review and Assessment of the Hybrid Techniques and Data Pre-Processing"

_hydrology, doi:10.3390/hydrology9070115_

Round 1
Reviewer 1 Report
General comment.
Although this is supposed to be a review, at this stage it seems like a collection of different information put together with no reasonable connection and lack of description. The authors have to alter the presentation of the subject, provide more information on methods, models etc and make it readable. You should address all comments below.
The word “drought” is repeated everywhere in the Introduction, it is also used to start different paragraphs. It also keeps on appearing in Section 2. Avoid using it so frequently, find a different way to start paragraphs and refer to it.
- Drought à State of the art or Theoretical consideration or ……..
If this Section cannot be embedded in the Introduction, do the opposite, take info from the Introduction to strengthen this Section.
4.3. Selecting the best model inputs: à Selecting appropriate descriptors (?)
When referring to specific techniques (e.g., data cleaning) you have to add more information. The way that research is presented is not inclusive, it certainly needs more info on data cleaning, normalization, it also needs relevant equations.
Moreover, there is a distinct region in ML called feature engineering. It covers a wide range of preprocessing methods. Since you review all trends in literature, you have to add more info on this. For example, what is dimensionality reduction? This is widely incorporated (see for example, Scientific Reports (2021) 11, 12520). Also, input weighting is another issue [Calphad (2020) 68, 101728].
A flow diagram (graphical) of the processes you describe in the first Sections is needed, it will become clearer to the reader.
Figure 1: Articales à You mean Articles?
The percentage number of studies à Percentage of studies
Zubaidi, et al. [73] mentioned that – re-phrase it
- Machine learning models. Maybe you mean Machine learning algorithms. You present only algorithms with very little information. Which is the selection criterion of these? Why haven’t you presented, for example, Gradient Boosting Regressor? Moreover, is ARIMA a ML technique? It is a different statistical model for time series analysis. Here you have to refer to ALL current ML methods used in relevant studies and why (see, for example, the presentation used in another review, [Fluids 7 (2022) 116]).
- Performance Metrics. The same as above. Since you make a review, you have to refer to more metrics used (e.g., AAD, AIC)
“Figure 2. A review of the hybrid model” needs more explanation. What is shown and why?
7.1. model-metaheuristic
Section title should change. Literature presentation in this Section should also change. It is not wise to start paragraphs with author names and just provide a glimpse of their work. Presentation should include a logical flow. You must refer to a method, describe it, and in the end of each sentence you can add the reference (ideally, without author names). You should also do the same for the next subsections:
7.2. Model with pre-processing
7.3. Hybrid with preprocessing
Also, based on the literature review, this study recommends that further research should focus on the following areas: - re-phrase
The first paragraph in Conclusions should not be here, it is repeating Introduction issues.
Author Response
Reviewer #1 comment:
Although this is supposed to be a review, at this stage it seems like a collection of different information put together with no reasonable connection and lack of description. The authors have to alter the presentation of the subject, provide more information on methods, models etc and make it readable. You should address all comments below.
Authors’ comment: Thank you very much for your feedback. All these issues are amended as below. Also, the language has now been thoroughly revised. We believe the paper now is much better. (all amended items are highlighted).
- The word “drought” is repeated everywhere in the Introduction, it is also used to start different paragraphs. It also keeps on appearing in Section 2. Avoid using it so frequently, find a different way to start paragraphs and refer to it.
Authors’ comment: Agree, amendments have been made accordingly. Thanks.
- Drought à State of the art or Theoretical consideration or ……..If this Section cannot be embedded in the Introduction, do the opposite, take info from the Introduction to strengthen this Section.
Authors’ comment: We agree with this comment and have revised it accordingly by taking info from the introduction to strengthen drought section.
- 3. Selecting the best model inputs: à Selecting appropriate descriptors (?)
Authors’ comment: This has now been addressed.
When referring to specific techniques (e.g., data cleaning) you have to add more information. The way that research is presented is not inclusive, it certainly needs more info on data cleaning, normalization, it also needs relevant equations.
Authors’ comment: This has now been addressed in the current version of the manuscript in sections 5.1 and 5.2.
Moreover, there is a distinct region in ML called feature engineering. It covers a wide range of preprocessing methods. Since you review all trends in literature, you have to add more info on this. For example, what is dimensionality reduction? This is widely incorporated (see for example, Scientific Reports (2021) 11, 12520). Also, input weighting is another issue [Calphad (2020) 68, 101728].
Authors’ comment: Amendments have been made accordingly, and the two valuable papers were considered in section 5.3.
- A flow diagram (graphical) of the processes you describe in the first Sections is needed, it will become clearer to the reader.
Authors’ comment: The amendment was implemented in Figure 2.
- Figure 1: Articales à You mean Articles?
Authors’ comment: This has now been addressed in Figure1.
- The percentage number of studies à Percentage of studies
Authors’ comment: This has now been addressed in Figure1.
- Zubaidi, et al. [73] mentioned that – re-phrase it
Authors’ comment: The sentence was amended to say “As a result, converting continuous variables is essential for creating time series that are normally or close to normally distributed [86]”. Please see the sentence in context on section 5.1, lines 189-190.
- Machine learning models. Maybe you mean Machine learning algorithms. You present only algorithms with very little information. Which is the selection criterion of these? Why haven’t you presented, for example, Gradient Boosting Regressor? Moreover, is ARIMA a ML technique? It is a different statistical model for time series analysis. Here you have to refer to ALL current ML methods used in relevant studies and why (see, for example, the presentation used in another review, [Fluids 7 (2022) 116]).
Authors’ comment: Thank you for pointing this out. We agree with this comment and have revised it accordingly. The amendment was implemented, and the valuable paper was considered in section in section 4.
- Performance Metrics. The same as above. Since you make a review, you have to refer to more metrics used (e.g., AAD, AIC)
Authors’ comment: Different metrics were added, including the suggested metrics (section 7).
- “Figure 2. A review of the hybrid model” needs more explanation. What is shown and why?
Authors’ comment: In the current version of the paper, we have clarified the methodology of the hybrid model of the study by using the same classifications in Hajirahimi and Khashei (2022). The edits related to this point can be found in:
- Introduction section 1, lines 82-97.
- Figure 3.
- Section 6.
We thank the reviewers for this suggestion, as we feel now that the methodology is much more clearly elaborated in the paper.
Hajirahimi, Z.; Khashei, M. Hybridization of hybrid structures for time series forecasting: a review. Artificial Intelligence Review 2022, 10.1007/s10462-022-10199-0, doi:10.1007/s10462-022-10199-0.
- 1. model-metaheuristic
Section title should change. Literature presentation in this Section should also change. It is not wise to start paragraphs with author names and just provide a glimpse of their work. Presentation should include a logical flow. You must refer to a method, describe it, and in the end of each sentence you can add the reference (ideally, without author names). You should also do the same for the next subsections:
7.2. Model with pre-processing
7.3. Hybrid with preprocessing
Authors’ comment: This has now been addressed. Please see the answer for point No. 10. (the above point).
- Also, based on the literature review, this study recommends that further research should focus on the following areas: - re-phrase
Authors’ comment: The sentence was amended to say “Also, based on analysis of the previous studies in this field, hybrid models can be improved by the following”. Please see the sentence in context on section 8 lines 475-476.
- The first paragraph in Conclusions should not be here, it is repeating Introduction issues.
Authors’ comment: This has now been addressed. Thanks.
Reviewer 2 Report
This study provides a literature review on the machine learning and hybrid methods employed to forecast droughts. The authors have conducted meticulous work to report the findings of a series of recent publications regarding this important issue. However, the manuscript does not meet the quality criteria to be published in Hydrology. The issues that need to be addressed are described below.
The manuscript has inappropriate structure.
The Abstract section contains many unnecessary details and self-evident statements. In fact, the second half of the Abstract section (lines 25 - 32) does not provide any useful information.
The Introduction section contains an extended discussion about well-founded and self-evident facts. The first half of this section (up to line 77) describes climate change, the consequent drought intensification, and the resulting threats. Only the last two lines of the Introduction section are devoted to the description of the manuscript motivation/objective: "Therefore, in addition to the previous literature review, this paper focuses on data pre-processing, data-driven modelling strategies) i.e., Univariate or Multivariate), machine learning models, performance metrics, and hybrid models."
Section 2. Drought: A whole section is dedicated to discussing once again issues previously discussed in the Introduction section (droughts predictability, impacts, indexes).
The manuscript has vague sentences and lacks details.
Figure 1. This figure gives "The percentage number of studies on hybrid ML models used to predict drought indices over the years." It is not mentioned what is the reference value with which the studies on hybrid ML models are compared (what is the denominator of the percentage value). It is not described how the displayed percentage values were obtained.
Table 1 provides the classification of drought based on SPI. The manuscript mentions many other indexes (SHDI, SIAP, SPEI, RDI, PDSI, etc.) for which the only information provided is a reference to a publication. Why is this table provided for SPI, whereas no further information for the rest of the indexes? A section devoted to these indexes will be greatly appreciated.
"When the data-driven modelling strategy is univariate, different methods are used to choose the best input variables, ..." Since it is univariate, a single variable was expected to be chosen.
"The number of predictors is important to create a perfect model to determine the appropriate sample size, as shown in Equation (2) [19]." The meaning is not clear, especially the meaning of "perfect model".
Figure 2. The concept behind this classification is not clear. The higher node is "Hybrid Model". Underneath of it there are three nodes "Model-Metahurstic" (?), "Preprocessing model", and "Hybrid with preprocessing". This structure results in an illogical classification that includes the following classes:
"Hybrid Model using Model-Metahurstic",
"Hybrid Model using Preprocessing model", and
"Hybrid Model using Hybrid with preprocessing".
Author Response
Reviewer #2 comment:
This study provides a literature review on the machine learning and hybrid methods employed to forecast droughts. The authors have conducted meticulous work to report the findings of a series of recent publications regarding this important issue. However, the manuscript does not meet the quality criteria to be published in Hydrology. The issues that need to be addressed are described below.
Authors’ comment: Thank you very much for your feedback. All these issues are amended as below. Also, the language has now been thoroughly revised. We believe the paper now is much better. (all amended items are highlighted).
- The manuscript has inappropriate structure.
Authors’ comment: In the current version of the paper, we have made a stronger effort to amended this issue. Thanks.
- The Abstract section contains many unnecessary details and self-evident statements. In fact, the second half of the Abstract section (lines 25 - 32) does not provide any useful information.
Authors’ comment: Thank you for pointing this out. The selected lines have now been thoroughly revised.
- The Introduction section contains an extended discussion about well-founded and self-evident facts. The first half of this section (up to line 77) describes climate change, the consequent drought intensification, and the resulting threats. Only the last two lines of the Introduction section are devoted to the description of the manuscriptmotivation/ objective: "Therefore, in addition to the previous literature review, this paper focuses on data pre-processing, data-driven modelling strategies) i.e., Univariate or Multivariate), machine learning models, performance metrics, and hybrid models."
Authors’ comment: This has now been addressed in the current version of the manuscript.
- Section 2. Drought: A whole section is dedicated to discussing once again issues previously discussed in the Introduction section (droughts predictability, impacts, indexes).
Authors’ comment: Please see the Authors’ comment for reviewer 1 , point No. 2.
- The manuscript has vague sentences and lacks details.
Authors’ comment: In the current version of the paper, we have made a stronger effort to address this issue.
- Figure 1. This figure gives "The percentage number of studies on hybrid ML models used to predict drought indices over the years." It is not mentioned what is the reference value with which the studies on hybrid ML models are compared (what is the denominator of the percentage value). It is not described how the displayed percentage values were obtained.
Authors’ comment: Amendments have been made accordingly.
- Table 1 provides the classification of drought based on SPI. The manuscript mentions many other indexes (SHDI, SIAP, SPEI, RDI, PDSI, etc.) for which the only information provided is a reference to a publication. Why is this table provided for SPI, whereas no further information for the rest of the indexes? A section devoted to these indexes will be greatly appreciated.
Authors’ comment: We have taken this advice, and the current version has four tables (Tables 1, 2, 3, and 4, Section 2).
- "When the data-driven modelling strategy is univariate, different methods are used to choose the best input variables, ..." Since it is univariate, a single variable was expected to be chosen.
Authors’ comment: The sentence was amended to say “When the data-driven modelling strategy is univariate, different methods are used to choose the best antecedent lags scenario, e.g., the Mutual Information technique”. Please see the sentence in context on section 5.3 lines 220-221.
- "The number of predictors is important to create a perfect model to determine the appropriate sample size, as shown in Equation (2) [19]." The meaning is not clear, especially the meaning of "perfect model".
Authors’ comment: The sentence was amended to say “According to Tabachnick and Fidell [84], the required sample size for prediction model is based on the number of predictors (independent variables) as shown in Equation (5) (e.g., if the number of predictors is four variables, the model needs 104+4=108 cases)”. Please see the sentence in context on section 5.3 lines 229-231.
- Figure 2. The concept behind this classification is not clear. The higher node is "Hybrid Model". Underneath of it there are three nodes "Model-Metahurstic" (?), "Preprocessing model", and "Hybrid with preprocessing". This structure results in an illogical classification that includes the following classes:
"Hybrid Model using Model-Metahurstic",
"Hybrid Model using Preprocessing model", and
"Hybrid Model using Hybrid with preprocessing".
Authors’ comment: Please see the Authors’ comment for reviewer 1 , point No. 10. Thanks.

Round 2
Reviewer 1 Report
I believe that the manuscript has been successfully revised and can be published at this form.
Author Response
Reviewer #1 comment:
I believe that the manuscript has been successfully revised and can be published at this form.
Authors’ comment: Many thanks for your positive opinion about the manuscript.

Reviewer 2 Report
GENERAL COMMENTS
The manuscript has improved, but the text still needs polishing. Please see specific comments. Also, an important study regarding the stochastic forecasting of droughts should be included in the literature review (after line 73). Please add the text: "AI has also been used for forecasting droughts by employing a multilayer perceptron network for stochastic synthesis of daily rainfall time series that preserve the Hurst coefficient and the autocovariance structure of all time scales [1]. This statistical property is very important in water management applications because it is related to “the tendency of wet years to cluster into multi-year wet periods or of dry years to cluster into multi-year drought periods” [2]".
SPECIFIC COMMENTS
Location: "Combining various prediction methods to create efficient hybrid models has become the most popular in recent years."
Comment: The most popular what?
Location: "In addition, using statistical criteria is essential to evaluate the performance of the models, such as RMSE, MAE, NSE, MPE, SI, BIC, AIC, and AAD."
Comment: -> "In addition, using statistical criteria, such as RMSE, MAE, NSE, MPE, SI, BIC, AIC, and AAD, is essential to evaluate the performance of the models."
Comment: "MPE", "SI", "AIC", and "AAD" are not included in the list of abbreviations.
Location: "Drought has become more common all over the world ... As a result, it is responsible for many natural disasters that occur in the world, that is the least understood natural disasters, such as drought-related economic losses in the United States alone, are estimated to be six to eight billion dollars per year [3]."
Comment: The syntax of these lines is not clear, and they have very vague meaning. Throughout history, humans have usually viewed droughts as "disasters" due to the impact on food availability and the rest of society. Humans have often tried to explain droughts as either a natural disaster, caused by humans, or ... (from Wikipedia). Droughts ARE natural disasters per se, they ARE NOT "responsible for" natural disasters.
Location: " As a result, the complex character and non-linear of the drought process require ..."
Comment: -> "the complex character and non-linearity"
Location: "Recently, hybrid models have become more popular hydrology prediction models."
Comment: -> "Recently, hybrid models have become more popular than hydrologic models."
Location: "The primary idea behind the OBH models is to use optimisation algorithms to describe the training process and identify the optimal parameters of the models [36]."
Comment: -> "The primary idea behind the OBH models is to use optimisation algorithms to identify the optimal parameters of the models [36]."
Location: "... optimal parameters of the models [36]. combine the models with ...
Comment: The syntax is not correct. Either start a new sentence or the period should be replaced with a comma.
Location: "... machine learning algorithms (i.e., advantages and ..."
Comment: This parenthesis does not close.
Location: 2. Drought a state of the art
Comment: This heading makes no sense.
REFERENCES
1. Rozos, E.; Dimitriadis, P.; Bellos, V. Machine Learning in Assessing the Performance of Hydrological Models. Hydrology 2022, 8, 67.
2. Koutsoyiannis, D. Hydrologic persistence and the Hurst phenomenon. In Water Encyclopedia, Vol. 4, Surface and Agricultural Water; Lehr, J.H., Keeley, J.W., Lehr, J.K., Kingery, T.B., Eds.; John Wiley & Sons: Hoboken, NJ, USA, 2005; Chapter 1.
Author Response
Reviewer #2 comment:
The manuscript has improved, but the text still needs polishing. Please see specific comments. Also, an important study regarding the stochastic forecasting of droughts should be included in the literature review (after line 73). Please add the text: "AI has also been used for forecasting droughts by employing a multilayer perceptron network for stochastic synthesis of daily rainfall time series that preserve the Hurst coefficient and the autocovariance structure of all time scales [1]. This statistical property is very important in water management applications because it is related to “the tendency of wet years to cluster into multi-year wet periods or of dry years to cluster into multi-year drought periods” [2]".
Authors’ comment: Thank you very much for your feedback. All these issues are amended as below. Also, the language has now been thoroughly polished. We believe the paper now is much better. (all amended items are highlighted).
- Amendments have been made accordingly, and the two valuable papers were considered in section 1, lines 81-86.
SPECIFIC COMMENTS
- Location: "Combining various prediction methods to create efficient hybrid models has become the most popular in recent years."
Comment: The most popular what?
Authors’ comment: The sentence was amended to say “. Combining various prediction methods to create efficient hybrid models has become the most popular use in recent years”. Please see the sentence in context on abstract section, lines 23-24.
- Location: "In addition, using statistical criteria is essential to evaluate the performance of the models, such as RMSE, MAE, NSE, MPE, SI, BIC, AIC, and AAD."
Comment: -> "In addition, using statistical criteria, such as RMSE, MAE, NSE, MPE, SI, BIC, AIC, and AAD, is essential to evaluate the performance of the models."
Comment: "MPE", "SI", "AIC", and "AAD" are not included in the list of abbreviations.
Authors’ comment: This has now been addressed. (abstract section , lines 27-29, and abbreviations section).
- Location: "Drought has become more common all over the world ... As a result, it is responsible for many natural disasters that occur in the world, that is the least understood natural disasters, such as drought-related economic losses in the United States alone, are estimated to be six to eight billion dollars per year [3]."
Comment: The syntax of these lines is not clear, and they have very vague meaning. Throughout history, humans have usually viewed droughts as "disasters" due to the impact on food availability and the rest of society. Humans have often tried to explain droughts as either a natural disaster, caused by humans, or ... (from Wikipedia). Droughts ARE natural disasters per se, they ARE NOT "responsible for" natural disasters.
Authors’ comment: The sentence was amended to say “It is one of the most expensive and least understood natural disasters. It may have devastating consequences on agriculture, water supply, ecosystems, public health, watershed health, and the economy. Each year, the United States suffers losses of six to eight billion dollars as a result of droughts [3]”. Please see the sentence in context on section 1, lines 36-39.
- Location: " As a result, the complex character and non-linear of the drought process require ..."
Comment: -> "the complex character and non-linearity"
Authors’ comment: This has now been addressed. (section 1, line 74).
- Location: "Recently, hybrid models have become more popular hydrology prediction models."
Comment: -> "Recently, hybrid models have become more popular than hydrologic models."
Authors’ comment: The sentence was amended to say “Recently, hybrid prediction models have become more popular than standalone models in hydrology”. Please see the sentence in context on section 1, lines 91-92.
- Location: "The primary idea behind the OBH models is to use optimisation algorithms to describe the training process and identify the optimal parameters of the models [36]."
Comment: -> "The primary idea behind the OBH models is to use optimisation algorithms to identify the optimal parameters of the models [36]."
Authors’ comment: Agree, amendments have been made accordingly. (section 1, lines 108-110).
- Location: "... optimal parameters of the models [36]. combine the models with ...
Comment: The syntax is not correct. Either start a new sentence or the period should be replaced with a comma.
Authors’ comment: This has now been addressed by replacing the period with a comma. (section 1, line 110).
- Location: "... machine learning algorithms (i.e., advantages and ..."
Comment: This parenthesis does not close.
Authors’ comment: This has now been addressed. (section 1, line 128).
- Location: Drought a state of the art
Comment: This heading makes no sense.
Authors’ comment: This has now been addressed. (section 2, line 131). Thanks.
